# Preparation and Characterization of Multielement Composite Oxide Nanomaterials Containing Ce, Zr, Y, and Yb via Continuous Hydrothermal Flow Synthesis

**DOI:** 10.3390/mi15010154

**Published:** 2024-01-20

**Authors:** Qingyun Li, Zihua Wang, Xuezhong Wang

**Affiliations:** 1School of Materials and Environment, Beijing Institute of Technology, Zhuhai, Zhuhai 519088, China; 01010@bitzh.edu.cn; 2School of Chemistry and Chemical Engineering, South China University of Technology, Guangzhou 510641, China; 3Pharmaceutical and Crystallization Systems Engineering Group, Beijing Key Laboratory of Enze Biomass and Fine Chemicals, School of Chemical Engineering, Beijing Institute of Petrochemical Technology, Beijing 102617, China

**Keywords:** confined jet mixer, multielement composite oxide, continuous hydrothermal flow synthesis, oxygen storage capacity, ionic conductivity

## Abstract

The synthesis of multielement composite oxide nanomaterials containing Ce, Zr, Y, and Yb was investigated using a micro confined jet mixer reactor operated in continuous mode under supercritical water conditions. The obtained nanoparticles were characterized using ICP-AES, SEM-EDS, FTIR, Raman spectroscopy, XRD, and TEM. All samples exhibited a uniform particle shape and a narrow particle size distribution. An analysis of the d-spacing results using selected electron area diffraction (SAED) patterns confirmed the production of cubic-phase crystals. A BET test was employed to determine the specific surface area of the prepared nanoparticles. OSC and TPR techniques were utilized to characterize the oxygen storage capacity and reduction performance of the obtained samples, with an analysis conducted to determine how the different proportions of elements affected the performance of multielement mixed oxides. The ionic conductivity of multielement composite oxide was measured using alternating current impedance spectroscopy (EIS), and the impact of Y, Ce, and Yb on the electrolyte material’s ionic conductivity was analyzed.

## 1. Introduction

Multicomponent composite oxide has special structural characteristics, showing superior thermal properties, mechanical properties, magnetic properties, optical properties, surface activity, and kinetic properties compared to single oxide, and has a wide range of applications in catalysis, environmental protection, energy fields, and other areas.

In this paper, the four elements of Ce, Zr, Y, and Yb were selected to make composite oxides in different proportions. Compared with the oxides of single metals, the composite oxides of multiple metals often have more advantages in performance and application [1,2]. When a certain amount of Y^3+^ is introduced into a zirconia lattice to form YSZ, Y^3+^ cations replace Zr^4+^ in the lattice. In order to maintain local electrical neutrality, oxygen vacancies are generated in the zirconia lattice, which can reduce the repulsion between local O^2−^, cause large lattice distortion in the ligand space, and release part of the interlayer stress to stabilize the cubic-phase C-ZrO_2_ [3,4]. Zacate et al. [5] demonstrated that the incorporation of rare earth elements into zirconia resulted in the formation of association defects with oxygen vacancies, thereby establishing a defect association energy between them. The presence of this defect association energy significantly enhanced the binding force between rare earth elements and zirconia crystals. CeO_2_ has a fluorite structure and can be combined with ZrO_2_ to form cubic CeO_2_-ZrO_2_ composite oxide, which is a catalyst with important application value in the field of gas environmental protection. At the same time, the CeO_2_-ZrO_2_ composite oxide is also the most common oxygen storage material because the Ce^4+^/Ce^3+^ has a reversible conversion process [6,7]. In recent years, Yb has emerged and developed rapidly in the fields of optical fiber communication and laser technology. Yb_2_O_3_ is a critical material in many high-tech materials such as laser glass and fiber. The electrical conduction in Yb-doped BaZrO_3_ has been investigated by researchers [8]. Xu et al. [9] introduced association defects by doping the rare earth element Yb into YSZ. They speculated that the defect association between Yb, Y, and oxygen defects could make the crystal more stable, and the doped elements were not easy to remove from the crystal.

In addition, the preparation method has a significant impact on the structures and properties of the materials, such as the phase structure, pore size distribution, particle size, catalytic activity, oxygen storage capacity, and electrochemical performance [10]. Lei Yang et al. prepared the hybrid ion conductor BaZr_0.1_Ce_0.7_Y_0.2−x_Yb_x_O_3−d_, which allows the rapid transport of proton and oxide ion vacancies and exhibits high ionic conductivity at relatively low temperatures (500 °C to 700 °C) [2]. Liu et al. found that the partial replacement of Ce in BaCe_0.8_Y_0.2_O_3_ with 10 mol% Zr, namely the formation of BaCe_0.7_Zr_0.1_Y_0.2_O_3−δ_, could significantly enhance its chemical stability [11]. Zuo et al. reported the effect of Zr-doped BZCY_s_ (Ba(Zr_0.8−x_Ce_x_Y_0.2_)O_3−α_) (0.4 ≤ x ≤ 0.8) on proton conductivity and chemical stability. The newly prepared Ni-BZCY_7_ (Ni-Ba(Zr_0.1_Ce_0.7_Y_0.2_)O_3−α_) cermet composite membrane not only has high proton conductivity but also has sufficient stability in an atmosphere containing CO_2_ and H_2_O [12].

The traditional methods developed for producing Zr-based nanoparticles, including precipitation [13], hydrolysis [14], pyrolysis [15,16], sol–gel [17], citrate complex [18], and microwave [19] methods, involved high energy consumption in the post-treatment process. Another disadvantage of the traditional preparation methods was the use of organic solvents. The hydrothermal route has been promoted for the production of zirconia particles and related ceramics for many years. However, the traditional hydrothermal process with a batch reactor was time-consuming, which led to particles with diameters in the submicron range and generally produced agglomeration [20].

Continuous hydrothermal flow synthesis (CHFS) using supercritical water as a reaction medium has been exploited to synthesize metal oxide nanoparticles from metal aqueous salt solutions. Supercritical water (SCW) has peculiar physicochemical properties such as a low dielectric constant and low polarity, leading to a high hydrolytic reaction rate and low solubility of metal oxides, which is preferable for terminating crystallization as early as possible after nucleation and contributes to the generation of ultra-fine and highly crystalline products. The CHFS process has attracted widespread attention because of its short reaction time (only a few seconds), moderate operation conditions, conveniently controlled process conditions, and simple post-processing. Even more remarkably, the nanoparticles produced via the CHFS process are ultra-fine and have a narrow particle size distribution (PSD).

An effective way to enhance solution mixing in a CHFS system is to design and configure a proper synthesis reactor. The first ever continuous micro T-shaped mixing reactor used in hydrothermal synthesis was introduced by T. Adschiri [21]. The precursor solution and hot water were premixed at a T mixing point and fed into a vertical stainless-steel tube. The tube served as a heater to maintain the solution at a high temperature for a while. It is believed that a fast hydrothermal reaction happened at the mixing point [22]. Sue et al. adopted a modified T-mixer design with different diameters for the two inlet pipes [23,24]. Blood et al. proposed a different microreactor design called a ‘counter-current reactor’ [25,26]. This type of reactor uses a counter-current configuration for the inlet streams. The SCW flows downwards through a thin tube while a precursor solution is fed upwards, and the two mix with each other at a ‘nozzle’ structure joint point. In this work, a micro confined jet mixer (CJM) reactor was employed in a CHFS system (Figure 1a). The micro confined jet mixer consisted of an inner tube inserting upwards into a larger outer tube (Figure 1b). Superheated water flowed upward in the inner tube to the mixing point, where it mixed with a metal salt solution, which also flowed upward from the two precursor streams. The resulting product stream then flowed upward, leaving the mixer and entering the tubular heat exchanger. CFD simulation results show that the mixing in a confined jet mixer is much faster than in a counter-current reactor, and particles produced using a confined jet mixer have a smaller size and a uniform size distribution compared to those produced using a counter-current reactor [27,28]. 

In the current study, four kinds of composite oxides consisting of Zr, Y, Ce, and Yb were prepared via the CHFS method. The effects of different cations and proportions on the structures and properties of Zr-based composite oxides were investigated, and the morphologies of the prepared nanoparticles were characterized via inductively coupled plasma atomic emission spectroscopy (ICP-AES), scanning electron microscopy and energy-dispersive spectrometry (SEM-EDS), X-ray diffraction (XRD), transmittance electron microscopy (TEM), Fourier transform infrared spectroscopy (FTIR), and Raman spectroscopy (RAMAN). The specific surface areas of the prepared composite oxide nanoparticles were analyzed via a BET test. The oxygen storage and reduction properties of the samples were characterized by means of an oxygen storage capacity (OSC) test and a H_2_ temperature-programmed reduction test (H_2_-TPR). The ionic conductivity of the prepared samples was measured via electrochemical impedance spectroscopy (EIS), and the influence of the Y, Ce, and Yb elements on the ionic conductivity of the electrolyte materials was analyzed.

## 2. Materials and Methods

### 2.1. CHFS System and Microreactor

The experimental apparatus is shown in Figure 1a, and the details of the confined jet mixer (CJM) reactor are shown in Figure 1b. Water was pumped through an electrical preheating coil and heated to the appropriate temperature (723 K). Flow rates of 10 mL min^−1^ were used for the metal salt solution, base solution, and water stream. The system’s pressure was maintained at 24.1 MPa [29]. The obtained nanoparticles travelled upwards for rapid cooling. Then, the collected aqueous suspension was separated via centrifugation and washed in distilled water. The obtained nanoparticles were then dried in an oven at 343 K for 24 h.

### 2.2. Materials

Pure zirconyl nitrate (ZrO(NO_3_)_2_·xH_2_O (≥99.5%)), yttrium nitrate (Y(NO_3_)_3_·6H_2_O (≥99.99%)), cerous nitrate (CeN_3_O_9_·6H_2_O (≥99.95%)), and ytterbium nitrate (YbN_3_O_9_·5H_2_O (99.9%)) were used as starting materials. The starting solutions were prepared by dissolving metal salts in distilled water, and the preparation ratios of the precursor salt solutions are listed in Table 1. The pH was adjusted by adding a KOH (99.9% purity) solution for all experimental runs. All the chemical reagents were obtained from Aladdin Bio-Chem Technology Co., Ltd. (Shanghai, China).

### 2.3. Characterization

#### 2.3.1. SEM-EDS

A Zesis Merlin scanning electron microscopy energy-dispersive spectrometer (SEM-EDS) (ZEISS, Oberkochen, Germany) was used to characterize the compositions and contents of the elements on the surfaces of the materials. A powdered sample was dipped directly onto the sample table with a conductive adhesive and sprayed with gold prior to the test.

#### 2.3.2. ICP-AES

In this paper, an ICP-AES (PerkinElmer Optima 8300, PerkinElmer, Waltham, MA, USA) characterization test was used to determine the contents of metal elements in the prepared nanoparticles, and the samples were nitrated using aqua regia.

#### 2.3.3. XRD

The crystal structures of the obtained mixed oxides were confirmed via powder X-ray diffraction (XRD) patterns using monochromic Cu Kα radiation (X’pert Powder, PANalytical, Almelo, The Netherlands) operating at 40 kV and 300 mA.

#### 2.3.4. FTIR

The FTIR spectra were recorded using a Nicolet IS50 Fourier transform infrared spectrometer (Thermo Fisher Scientific, Waltham, MA, USA). The infrared spectral range was 4000 to 400 cm^−1^ at a resolution of 1 cm^−1^ with a mirror velocity of 0.88 cm/s, leading to modulation frequencies in the range of 70–704 Hz.

#### 2.3.5. RAMAN

The Raman spectra were obtained at room temperature using an HJY LabRAM Aramis laser Raman spectrometer (Horiba Jobin-Yvon GmbH, Oberursel, Germany) with an Ar ion laser with an excitation wavelength of 514.5 nm. Backscattering geometry was adopted for measurement under the conditions of a laser power of 50 mW and a resolution of 2 cm^−1^.

#### 2.3.6. TEM-SEAD-EDS

The particle sizes and morphologies of the as-prepared samples were investigated using a transmission electron microscope (TEM; JEOL JEM-2100F, Tokyo, Japan), selected area electron diffraction (SAED), and energy-dispersive X-ray spectroscopy (EDS). The TEM specimens were prepared by dispersing the dried powder in acetone and dropping it on a microgrid.

#### 2.3.7. BET Specific Surface Area Test

The BET surface areas of the samples were measured via N_2_ adsorption using an ASAP2010 system (Micromeritics Inc., Norcross, GA, USA). The samples were degassed at 120 °C for 4 h in a vacuum, and N_2_ adsorption was carried out at 77 K.

#### 2.3.8. Oxygen Storage Capacity (OSC) Test

The OSC is the total amount of O_2_ that can be extracted from a sample at a preestablished temperature and partial pressure of the reducing agent H_2_ with TPD/TPR (AutoChem1 II 2920). The samples were reduced with pure H_2_ for 40 min at 550 °C and then purified with deoxygenated high-purity N_2_ at a flow rate of 40 mL/min until the samples dropped to 200 °C. After the baseline of the recorder straightened, oxygen was pulsed every 5 min at a constant temperature (flow rate: 30 mL/min) to calculate the amount of oxygen adsorbed.

#### 2.3.9. H_2_-TPR Test

The H_2_ temperature-programmed reduction test device was the same as in the OSC test. The samples were weighed and placed in a quartz reaction tube at 400 °C, pretreated in a N_2_ atmosphere for 30 min, cooled to room temperature, and switched to a N_2_ (90%)–H_2_ (10%) mixture with a flow rate of 40 mL/min. After the baseline was flat, a temperature-programmed reduction was carried out at a heating rate of 10 °C/min, and it was detected using a thermal conductivity detector (TCD).

#### 2.3.10. Electrochemical Impedance Spectroscopy

Electrochemical impedance spectroscopy (EIS) is a technique applied to study solid electrolytes. In this experiment, a Chenhua CHI 660E electrochemical analyzer was used with a closed tube furnace and an air atmosphere. The test temperature was 500–1000 °C, measured every 50 °C, and the heating rate was 3 °C·min^−1^. The impedance map was fitted using software Zsimpwin Version 3.21.

## 3. Results and Discussion

The SEM-EDS energy spectra (Figure 2) and ICP-AES were used to characterize and analyze the chemical compositions and metal element contents of the prepared samples. The four samples of °YZ1, CYZ2, CYbZ, and CYYbZ did not have impurity peaks, and the purity was very high. The C element was produced by the carbon element contained in the conductive adhesive during the preparation process. The stoichiometric ratios obtained via analysis of the metal elements in the four samples were consistent with the initial set ratio and slightly deviated from the initial set ratio. This deviation was caused by the different precipitation kinetics of the different precursors. Precipitation was generated in a short time (only a few seconds), and cations with greater solubility entered the oxide less (see Table 2).

Figure 3 shows the XRD patterns of the four obtained samples, in which the diffraction peaks of CYZ1 at 2θ = 30.119°, 34.918°, 50.212°, 59.673°, 62.617°, 73.745°, 81.669°, and 84.268° correspond to cubic-phase ZrO_2_ (ICSD: 647689) crystal faces. The diffraction peaks of samples CYZ2, CYbZ, and CYYbZ at 2θ = 28.82°, 33.4°, 47.957°, 56.919°, 59.699°, 70.16°, 77.554°, 79.966°, and 89.480° correspond to cubic-phase CeO_2_-ZrO_2_ (ICSD: 165044) crystal faces. The sample CYZ1 exhibited the highest Zr content, while the low Y and Ce contents can be considered as dopants in the ZrO_2_ crystal lattice, leading to a partial substitution of Zr^4+^ ions through isomorphic substitution. This conclusion is supported by the XRD results, which revealed that CYZ1 samples doped with small amounts of Y and Ce elements only exhibited characteristic diffraction peaks corresponding to cubic-phase ZrO_2_. The molar ratios of the Ce and Zr elements in the CYZ2, CYbZ, and CYYbZ samples were the same, which were 70% and 10%, respectively. According to the XRD results, the crystal structures of these three samples were similar, which was consistent with the crystal structure of cubic-phase CeO_2_-ZrO_2_. The absence of the characteristic peaks of the doped Y and Yb elements in the XRD pattern indicates that these metal ions dissolved into the crystal lattice and homomorphic substitution occurred. The XRD peak-to-peak pattern of all four samples is wide and the peak intensity is relatively low, which indicates the nanoparticles had a small grain size. 

The FTIR diagram of the four prepared samples is shown in Figure 4. As shown in the figure, the wide absorption peaks in the range of 3450 cm^−1^ and the absorption peaks observed near 1630 cm^−1^ of all samples belong to the bending vibration and stretching vibration of the water molecules contained in the samples [30,31], respectively. The absorption peak at about 1538 cm^−1^ is caused by the vibration of the Zr-OH bond [32]. The absorption peak observed near 1350 cm^−1^ is the stretching vibration peak of the Zr-O bond [33]. The absorption peaks occurring around 1080, 865, and 525 cm^−1^ are characteristic absorption peaks of cubic-phase crystals [34,35].

In order to further analyze the phase structure of the four samples, a Raman spectrum analysis was performed on the samples. As shown in Figure 5, the Raman spectra of the four samples all show a strong characteristic peak between 400 and 500 cm^−1^, which is the characteristic peak of cubic-phase crystals, indicating that the four samples were all cubic phase, which is consistent with the results of the XRD and FTIR spectra above. Among them, the characteristic peak of sample CYZ1 is shifted in the direction of a low wave number. A possible reason for this is that the content of the Zr element in the CYZ1 sample was relatively large and, therefore, the structures of the formed crystals were similar to ZrO_2_. Combined with the XRD pattern of CYZ1, it can be seen that both of them were cubic-phase ZrO_2_ crystal structures. The characteristic peaks of the Raman spectra of the CYZ2, CYbZ, and CYYbZ samples are roughly the same. The difference among the three samples is that the contents of the Y and Yb elements were different, but the theoretical contents of the Ce and Zr elements were the same. Combined with the XRD patterns, the crystal structures of the three samples were similar, all of which were cubic-phase CeO_2_-ZrO_2_ crystal structures.

The surface morphology and crystal structure details of the samples were observed via TEM, SAED, and HR-TEM, and the results are shown in Figure 6, where the numbers (A)–(D) represent CYZ1, CYZ2, CYbZ, and CYYbZ, respectively. It can be seen in Figure 6A–D that all samples had a uniform particle shape and a narrow particle size distribution. The small figures in Figure 6A–D are histograms of the particle size distributions. The particle size distribution (PSD) data were determined by manually measuring around 200 particles using ImageJ. The columnar PSD diagram reveals that the individual nanoparticle size in all four samples was approximately 5 nm. Figure 6a–d show detailed crystal profiles taken using high-resolution electron microscopy (HR-TEM). The majority of the grains in the figure exhibit parallel and evenly spaced lattice stripes without any lattice mismatch, indicating excellent crystallization of the samples, which is consistent with the SAED results shown in Figure 6(a1–d1), confirming the XRD results. The samples were all crystals with good crystallinity. According to the data of the ICSD 647689 standard card, 2θ = 30.119°, 34.918°, and 50.212° correspond to the (111), (002), and (022) crystal planes of the crystal, and their lattice spacings are d = 0.2965 nm, 0.2568 nm, and 0.1816 nm, respectively. The CYZ1 sample in Figure 6a has three major lattice fringes with lattice spacings of 0.2986 nm, 0.2663 nm, and 0.1823 nm, which match the interfacial distances of (111), (002), and (022) crystal faces, respectively. According to the data of the ICSD 165044 standard card, 2θ = 28.820°, 33.40°, and 47.957° correspond to the (111), (002), and (022) crystal planes of the crystal, and the lattice spacings are d = 0.3095 nm, 0.2681 nm, and 0.1896 nm, respectively. In Figure 6b, the CYZ2 sample has three main lattice fringes with lattice spacings of 0.3174 nm, 0.2687 nm, and 0.1737 nm, which match the interfacial distances of (111), (002), and (022) crystal faces, respectively. In Figure 6c, the CYbZ sample has three main lattice fringes with lattice spacings of 0.3161 nm, 0.2738 nm, and 0.1888 nm, which match the interfacial distances of (111), (002), and (022) crystal faces, respectively. The CYYbZ sample in Figure 6d has three major lattice stripes with lattice spacings of 0.3147 nm, 0.2743 nm, and 0.1914 nm, which match the interfacial distances of (111), (002), and (022), respectively.

The BET specific surface areas of the CYZ1, CYZ2, CYbZ, and CYYbZ samples are shown in Figure 7. The BET surface area, constant C, and regression coefficient data for the four samples are presented in Table 3. It can be observed, based on the tabulated data, that the values of C for all samples fall within the range of 50–200, consistent with typical oxide characteristics. Moreover, all regression coefficients exceed 0.999, indicating excellent agreement between the adsorption lines of the samples (at P/Po = 0.05~0.35) and the BET equation. These results demonstrate a close correspondence between the test data and the actual specific surface area. As can be seen in Table 3, all four samples had large specific surface areas.

The oxygen storage capacity of the samples was tested using oxygen pulse adsorption technology. The oxygen storage capacity (OSC) test results of the four samples are shown in Table 4. Among the four samples, CYZ2 had the highest oxygen storage of 545.946 μmol/g. The oxygen storage of the CYZ1, CYbZ, and CYYbZ samples was 300–400 μmol/g. The Ce content of the CYZ2, CYbZ, and CYYbZ samples was 70% (mol%), and the Zr content was 10% (mol%). The Y content of CYZ2 was the highest, followed by that of CYYbZ, while CYbZ did not contain the Y element, and the oxygen storage of the three samples gradually decreased. This indicates that the entry of Y^3+^ into the lattice produced a large number of oxygen vacancies, which increased the activity capacity of oxygen in the body phase, resulting in a high oxygen storage capacity in the solid solution. However, the addition of Yb^3+^ into the lattice resulted in a reduction in structural defects in the lattice, a reduction in the oxygen hole concentration, and a decrease in oxygen storage.

Figure 8 shows the H_2_-TPR test results. As can be seen in the figure, all four samples had three reduction peaks between 300 and 800 °C. The reduction peak near 300 °C corresponded to surface oxygen species adsorbed on the composite oxides. The reduction peaks between 400 and 500 °C belonged to ${\mathrm{CeO}}_{2}\to {\mathrm{Ce}}_{2}O_{3}$ reduction [36], which occurred near 300 °C, showing that the transformation of the Ce^4+^/Ce^3+^ activation energy was reduced. That is, due to ion doping, the degree of crystal lattice distortion increased, resulting in a large number of lattice oxygen defects, improving the mobility of oxygen ions, and this process promoted the conversion between Ce^4+^ and Ce^3+^. Four samples had a high-temperature reduction peak between 600 and 800 °C, indicating the reduction temperatures of the bulk oxygen of multicomponent composite oxides. The high-temperature reduction peak of CYZ1 was between 600 and 700 °C, and the strength was weak because the content of Ce was very slight. The high-temperature reduction peaks of CYZ2 and CYbZ were around 700 °C and moved towards the high-temperature direction, and the strength increased because the content of Ce increased. The high-temperature reduction peak of CYYbZ was around 750 °C with a higher peak intensity. The interaction of the Yb and Y elements could produce more oxygen defects inside the multicomponent composite oxide crystal, and the defect association could make the crystal more stable. Therefore, the reduction peak intensity increased and migrated to a higher temperature. 

The ionic conductivity of the CYZ1, CYZ2, CYbZ, and CYYbZ samples as solid fuel cell electrolyte materials was measured using the electrochemical impedance spectroscopy method. About 0.5 g of a powdered sample was first pressed at 50 MPa and then sintered at 1500 °C to make the sample dense. After adding a binder, the sintered sample was formed into a disc with a diameter of about 8 mm and a thickness of about 2 mm. Then, it was sintered at 800 °C for about 1 h, and electrodes were prepared via the direct welding of platinum paste.

Table 5 shows the conductivity values of the samples at different temperatures, and Figure 9 shows the Arrhenius curves of the sample conductivity at different temperatures. It can be seen that the conductivity of the samples increased uniformly with the increase in the test temperature, and the logarithm of the conductivity had a linear relationship with the temperature, which satisfied the Arrhenius relationship well. It can be seen in the table that at 1000 °C, the conductivity values of CYZ1 and CYZ2 were 0.006284 and 0.00264 S·cm^−1^, respectively, and the conductivity values of CYbZ and CYYbZ were 0.01129 and 0.01586 S·cm^−1^, respectively, indicating that the conductivity significantly improved after doping with Yb.

## 4. Conclusions

In this study, we developed a novel process for synthesizing Zr, Y, Ce, and Yb composite oxides using a continuous hydrothermal flow synthesis (CHFS) system with a confined jet mixer (CJM) reactor. SEM-EDS energy spectra and ICP-AES were used to characterize the chemical compositions and the contents of the metal elements of the prepared samples. No impurity peak of metal elements appeared in the four prepared samples, and the purity was very high. The molar ratios obtained via analysis of the metal elements were roughly consistent with the initial set stoichiometric ratios. By employing XRD, FTIR, and Raman analysis techniques, it was observed that all the prepared samples exhibited a cubic-phase crystal structure. Notably, the CYZ1 sample with the highest Zr content demonstrated consistency with the cubic-phase ZrO_2_ crystal structure. Additionally, three samples containing high Ce contents (CYZ2, CYbZ, and CYYbZ) conformed to the crystal structure of cubic-phase CeO_2_–ZrO_2_. TEM images revealed a uniform particle shape and a narrow size distribution across all samples. A high-resolution TEM analysis indicated parallel and equidistant lattice fringes in most grains without any lattice mismatch, confirming excellent crystallization of the samples.

The ionic conductivity of these four samples when used as solid fuel cell electrolyte materials was measured using the electrochemical impedance spectroscopy method. After fitting using Zsimpwin software, grain boundary resistance and electrical conductivity values were obtained at different temperatures. It was observed that both grain boundary resistance and total resistance gradually decreased while electrical conductivity increased with increasing temperature. The logarithm of electrical conductivity exhibited a linear relationship with temperature in accordance with the Arrhenius relation. Furthermore, doping with Yb significantly enhanced the electrical conductivity of the sample.

## Figures and Tables

**Figure 1 micromachines-15-00154-f001:**
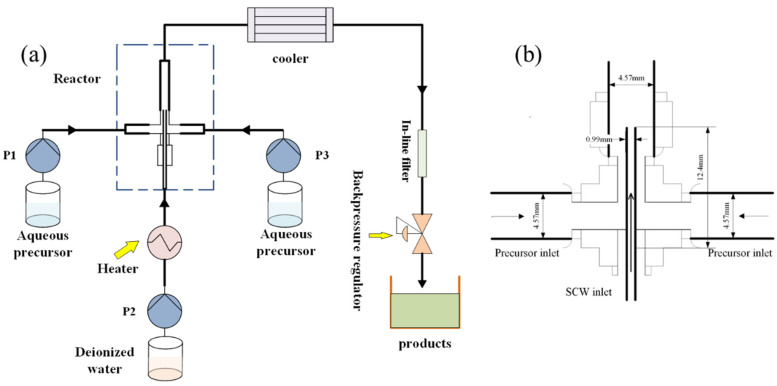
(**a**) Flow diagram of the CHFS system and (**b**) a schematic diagram of the micro CJM reactor.

**Figure 2 micromachines-15-00154-f002:**
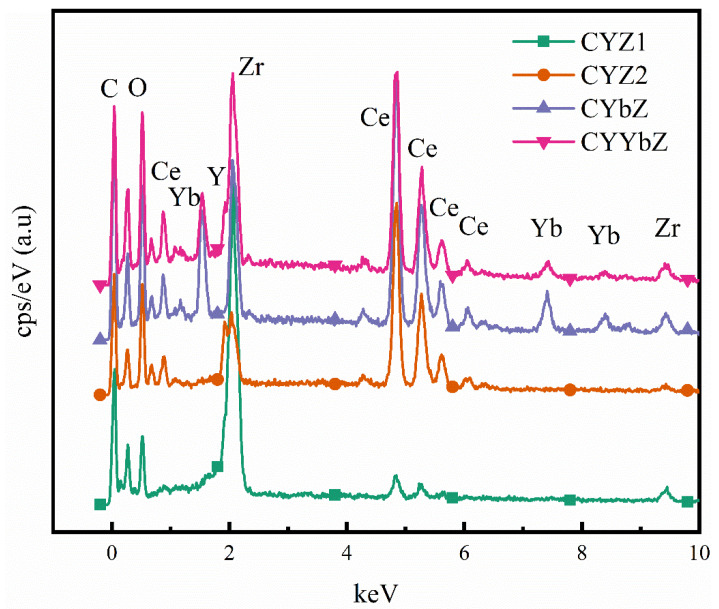
SEM-EDS spectra of the CYZ1, CYZ2, CYbZ, and CYYbZ samples.

**Figure 3 micromachines-15-00154-f003:**
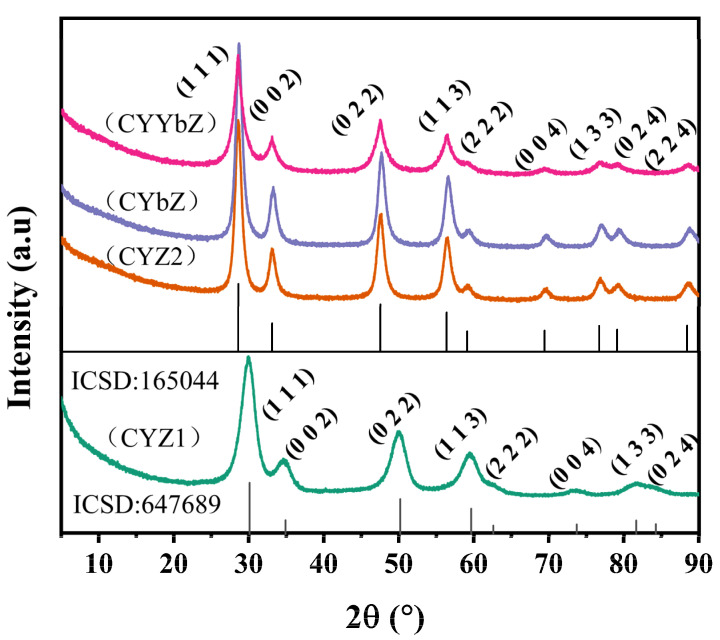
XRD analysis of the CYZ1, CYZ2, CYbZ, and CYYbZ samples.

**Figure 4 micromachines-15-00154-f004:**
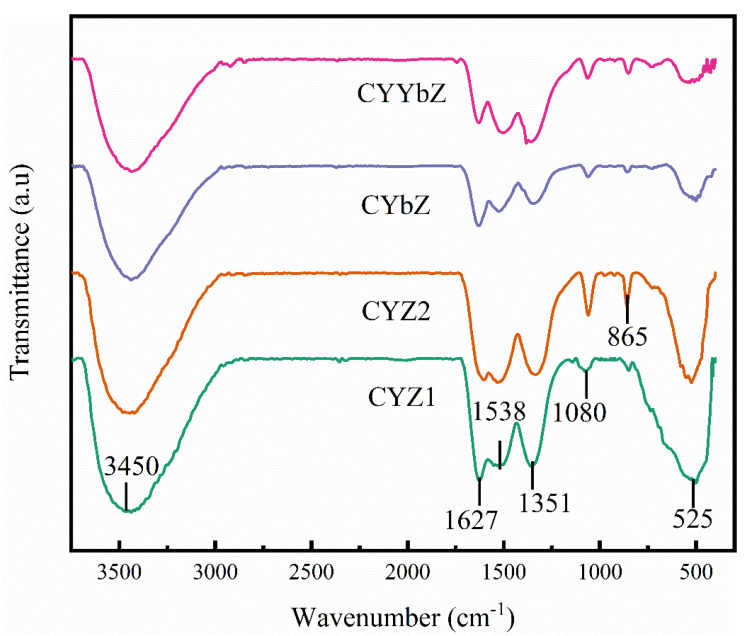
FTIR spectra of the CYZ1, CYZ2, CYbZ, and CYYbZ samples.

**Figure 5 micromachines-15-00154-f005:**
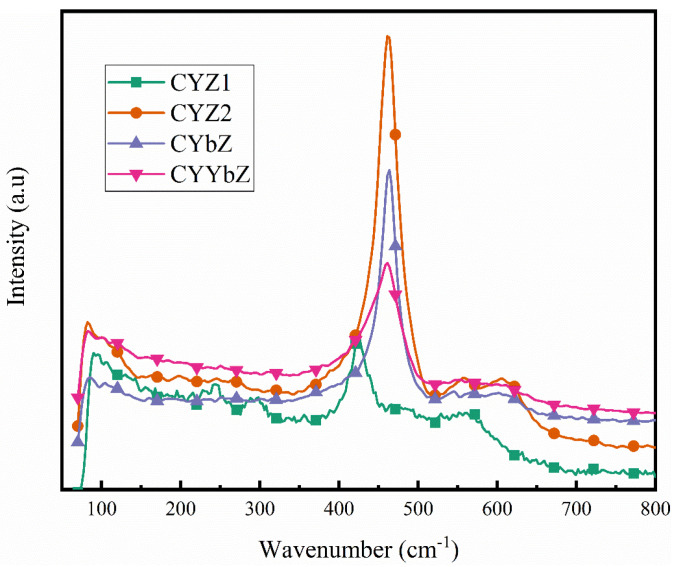
Raman spectra of CYZ1, CYZ2, CYbZ, and CYYbZ samples.

**Figure 6 micromachines-15-00154-f006:**
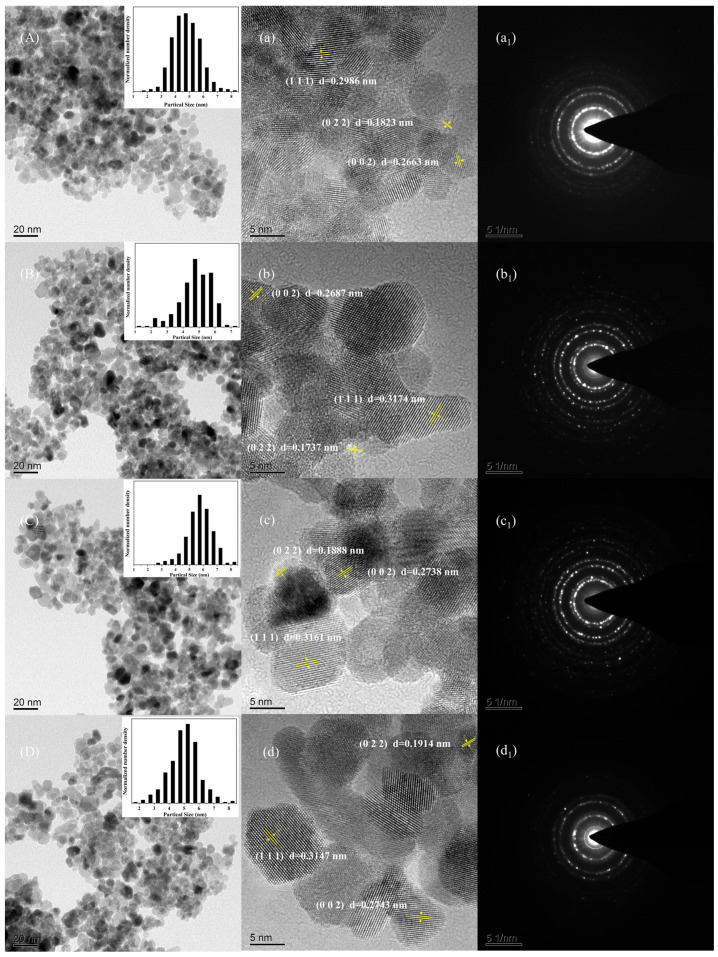
(**A**–**D**) TEM images, (**a**–**d**) HR-TEM images, and (**a_1_**–**d_1_**) SAED patterns of CYZ1, CYZ2, CYbZ, and CYYbZ samples.

**Figure 7 micromachines-15-00154-f007:**
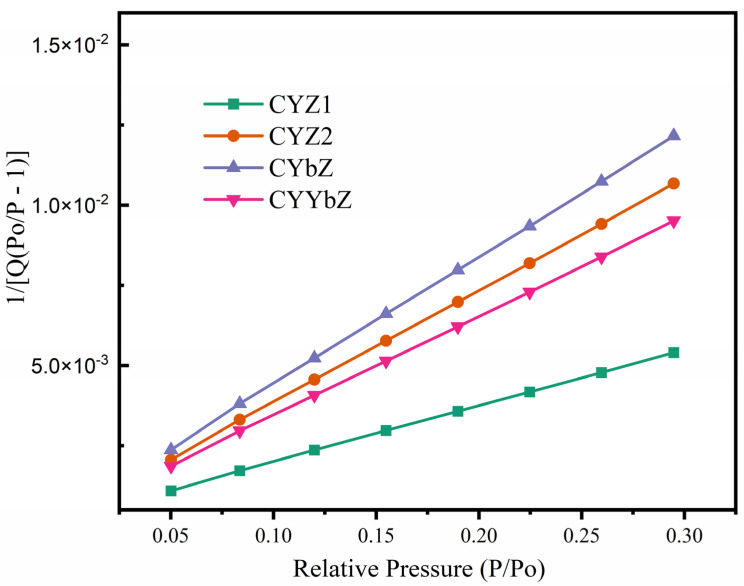
BET surface area plot of CYZ1, CYZ2, CYbZ, and CYYbZ samples.

**Figure 8 micromachines-15-00154-f008:**
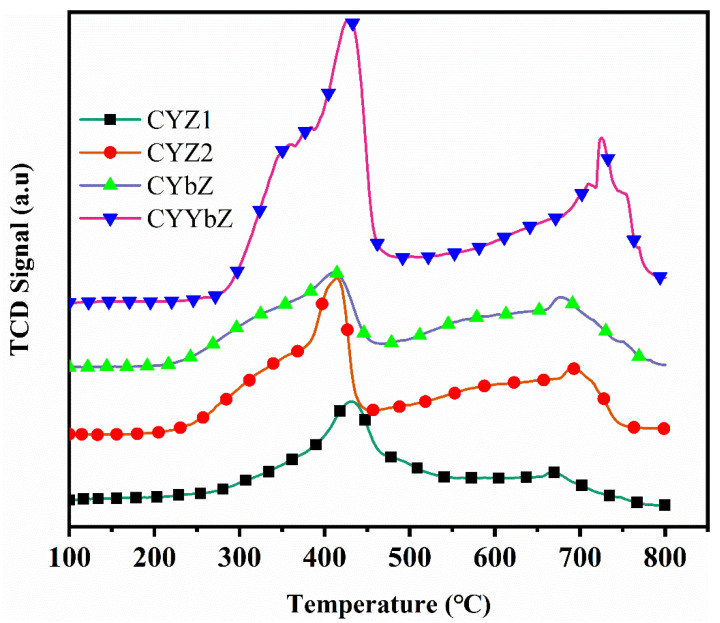
H_2_-TPR profiles of the CYZ1, CYZ2, CYbZ, and CYYbZ samples.

**Figure 9 micromachines-15-00154-f009:**
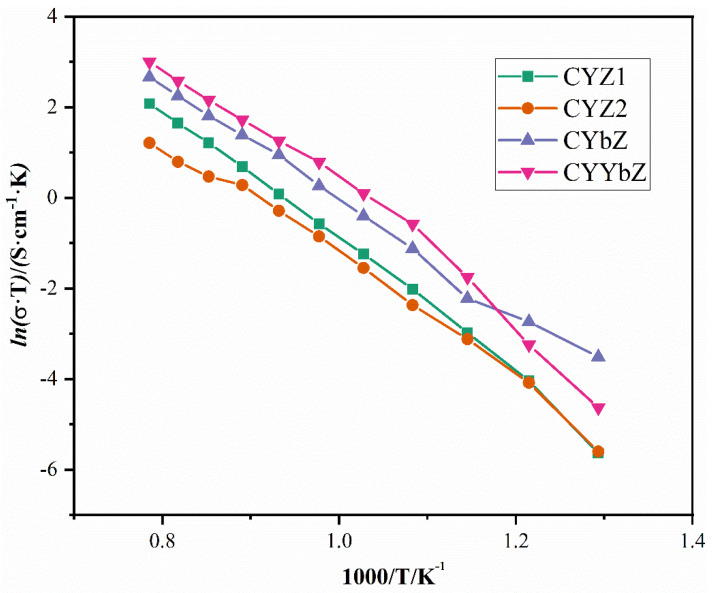
Arrhenius plots of CYZ1, CYZ2, CYbZ, and CYYbZ samples.

**Table 1 micromachines-15-00154-t001:** Zr-based multielement composite oxide precursor solution preparation table.

Salt Mass (g)	ZrO(NO_3_)_2_·x H_2_O	Y(NO_3_)_3_·6H_2_O	CeN_3_O_9_·6H_2_O	YbN_3_O_9_·5H_2_O	Molar Ratio
CYZ1	20.2326	2.8726	2.1711	------	Ce:Y:Zr = 5:7.5:87.5
CYZ2	2.3123	7.6602	30.3954	------	Ce:Y:Zr = 7:2:1
CYbZ	2.3123	-------	30.3954	8.9826	Ce:Yb:Zr = 7:2:1
CYYbZ	2.3123	3.8301	30.3954	4.4913	Ce:Y:Yb:Zr = 7:1:1:1

**Table 2 micromachines-15-00154-t002:** Composition analyses of the obtained CYZ1, CYZ2, CYbZ, and CYYbZ samples.

(Molar Ratio)	CYZ1 (Ce:Y:Zr)	CYZ2 (Ce:Y:Zr)	CYbZ (Ce:Yb:Zr)	CYYbZ (Ce:Y:Yb:Zr)
EDS	0.94:1.26:14.69	14.43:3.24:1.98	18.02:5.04:2.71	15.76:1.92:2.28:2.13
ICP-AES	1.18:1.53:16.07	15.39:3.76:2.24	16.81:4.91:2.53	18.73:2.19:2.71:2.59

**Table 3 micromachines-15-00154-t003:** BET data of CYZ1, CYZ2, CYbZ, and CYYbZ samples.

Data	CYZ1	CYZ2	CYbZ	CYYbZ
BET surface area (m^2^/g)	145.26	123.04	108.33	138.69
C	71.87	108.73	95.26	97.86
Correlation coefficient	0.999	0.999	0.999	0.999

**Table 4 micromachines-15-00154-t004:** OSC data of CYZ1, CYZ2, CYbZ, and CYYbZ samples.

Samples	CYZ1	CYZ2	CYbZ	CYYbZ
OSC (μmol/g)	306.81	545.95	331.87	381.06

**Table 5 micromachines-15-00154-t005:** Conductivity values of CYZ1, CYZ2, CYbZ, and CYYbZ at different temperatures.

T (°C)	CYZ1 σ/S·cm^−1^	CYZ2 σ/S·cm^−1^	CYbZ σ/S·cm^−1^	CYYbZ σ/S·cm^−1^
500	4.6279 × 10^−6^	4.7817 × 10^−6^	3.8585 × 10^−5^	1.2582 × 10^−5^
550	2.1348 × 10^−5^	2.0508 × 10^−5^	7.9124 × 10^−5^	4.7505 × 10^−5^
600	5.8468 × 10^−5^	5.0582 × 10^−5^	0.000124	0.0001987
650	0.0001445	0.0001018	0.0003543	0.0006076
700	0.0002970	0.0002185	0.0006911	0.001128
750	0.0005524	0.0004179	0.001279	0.002157
800	0.001010	0.0006999	0.002413	0.003267
850	0.001779	0.001181	0.003569	0.004974
900	0.002874	0.001365	0.005226	0.007369
950	0.004248	0.001812	0.007753	0.01080
1000	0.006284	0.002640	0.01129	0.01586

## Data Availability

Data are contained within the article.

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
