# Peer review of "Preparation and Characterization of Multielement Composite Oxide Nanomaterials Containing Ce, Zr, Y, and Yb via Continuous Hydrothermal Flow Synthesis"

_micromachines, 2024, doi:10.3390/mi15010154_

Round 1

Reviewer 1 Report

Comments and Suggestions for Authors

The submitted manuscript deals with the problem of utilizability of the Continuous Hydrothermal Flow Synthesis (CHFS) method for preparing Zr-based materials for possible utilization as solid electrolytes in electrochemical systems. The method used is not original. In their previous article, the authors published its utilization in synthesizing ZrO2/CeO2 mixed oxides (Nanomaterials 2022, 12, 668). The novelty of this study can be ascribed to the incorporation of other metals (Y, Yb) into the structure of the mixed oxides ZrO2/CeO2 to improve their electrical conductivity. However, the manuscript should be improved in some of its parts. The main problems are pointed out here:

1. The title of the manuscript is misleading as the basic oxide in three samples from four synthesized is CeO2 and not ZrO2.

2. Discussion and presentation of the XRD spectra are not adequate. The authors used non-standard symbols for angle degrees, and some of the values discussed in the text are missed in the spectra. It is interesting that not any part of the spectra reveals the presence of the other elements, only Zr and Ce are presented.

2. The authors presented TEM pictures of the prepared oxides, and therefore, it will be appropriate to also give the size distribution of the prepared oxide nanoparticles.

4. The values of the determined specific surface areas (Table 2) and oxygen capacity (Table 3) cannot be measured with such high precision (referred accuracy is better than 10-3 %).

5. Interpretation of the H2-TPR spectra should be improved.

6. The preparation of the sample for measurement of its conductivity is not described clearly.

Based on the comments above, I can recommend the publication of the submitted manuscript in the Micromachines journal after minor revision.

Author Response

Point 1: The title of the manuscript is misleading as the basic oxide in three samples from four synthesized is CeO2 and not ZrO2.

Response 1: Thank you for your valuable comment.

In the revised paper, the title of the article has been changed to “Preparation and characterization of multielement composite oxide nanomaterials containing Ce, Zr, Y, and Yb by Continuous Hydrothermal Flow Synthesis” according to your suggestion.

Point 2: Discussion and presentation of the XRD spectra are not adequate. The authors used non-standard symbols for angle degrees, and some of the values discussed in the text are missed in the spectra. It is interesting that not any part of the spectra reveals the presence of the other elements, only Zr and Ce are presented.

Response 2: 1) Non-standard symbols for angle degrees have been revised.

2) All values discussed in the text are marked in the spectrum. The diffraction peaks of CYZ1 at 2θ =30.119º, 34.918º, 50.212º, 59.673º, 62.617º, 73.745º, 81.669º and 84.268º correspond to eight lattice planes. The diffraction peaks of samples CYZ2, CYbZ and CYYbZ at 2θ =28.82º, 33.4º, 47.957º, 56.919º, 59.699º, 70.16º, 77.554º, 79.966 º and 89.480º correspond to nine lattice planes. These lattice plane parameters are all marked in Figure 3.

3) In response to your proposed “not any part of the spectra reveals the presence of the other elements, only Zr and Ce are presented”, we give the following explanation in the revised paper.

  The sample CYZ1 exhibits the highest Zr content, while the low Y and Ce contents can be considered as dopants in the ZrO2 crystal lattice, leading to partial substitution of Zr4+ ions through isomorphic substitution. This conclusion is supported by XRD results, which reveal that CYZ1 samples doped with small amounts of Y and Ce elements only exhibit characteristic diffraction peaks corresponding to cubic phase ZrO2. The molar ratios of Ce and Zr elements in CYZ2, CYbZ and CYYbZ samples are the same, which are 70% and 10%, respectively. According to the XRD results, the crystal structures of these three samples are similar, which is consistent with the crystal structure of cubic phase CeO2-ZrO2. The absence of characteristic peaks of the doped Y and Yb elements in the XRD pattern indicates that these metal ions are dissolved into the crystal lattice and homomorphic substitution occurs.( page 6, line220-230)

Point 3: The authors presented TEM pictures of the prepared oxides, and therefore, it will be appropriate to also give the size distribution of the prepared oxide nanoparticles.

Response 3: Thank you for your valuable comment.

We added the particle size distribution plot to the TEM pictures, and the corresponding elaboration is given in the text.

The small figure in Figure 6 (A)-(D) is the histogram of sample particle size distribution. The particle size distribution (PSD) data were determined by manually measuring around 200 particles using ImageJ. The columnar PSD diagram reveals that the individual nanoparticle size in all four samples is approximately 5nm. ( page 9, line266-269)

Point 4: The values of the determined specific surface areas (Table 2) and oxygen capacity (Table 3) cannot be measured with such high precision (referred accuracy is better than 10-3 %).

Response 4: This error has been revised. In the revised paper, the data in Tables 3 and 4 are both accurate to 0.01.

Point 5: Interpretation of the H2-TPR spectra should be improved.

Response 5: Interpretation of the H2-TPR spectra was improved as follows.

Figure 8 showed the H2-TPR test results. As can be seen from the figure, all four samples have three reduction peaks between 300-800ºC. The reduction peak about 300ºC is corresponding to surface oxygen species adsorbed on composite oxides. The reduction peaks Between 400-500ºC belong to reduction36,which is near 300℃, showed that transformation of Ce4+ / Ce3+ activation energy is reduced. That is, due to ion doping, the degree of crystal lattice distortion increases, resulting in a large number of lattice oxygen defects, improving the mobility of oxygen ions, and this process promotes the conversion between Ce4+ and Ce3+.Four samples have a high temperature reduction peak between 600-800 ºC indicated the reduction temperatures of bulk oxygen of multicomponent composite oxides. The high temperature reduction peak of CYZ1 is between 600-700℃, and the strength is weak because of the content of Ce is very slight. The high temperature reduction peak of CYZ2, CYbZ is around 700 ºC, moves towards high temperature direction, and the strength increases, because the content of Ce increases. The high temperature reduction peak of CYYbZ is around 750 ºC with higher peak intensity. The interaction of Yb and Y elements can produce more oxygen defects inside the multicomponent composite oxide crystal, and the defect association can make the crystal more stable. Therefore, the reduction peak intensity increases and migrates to high temperature. ( page 12, line323-340)

Point 6:  The preparation of the sample for measurement of its conductivity is not described clearly.

 Response 6: The description of the preparation of the sample for measurement of its conductivity has been revised as follows.

The ionic conductivity of CYZ1, CYZ2, CYbZ and CYYbZ samples as solid fuel cell electrolyte materials was measured by electrochemical impedance spectroscopy method. About 0.5g of the powdered sample was first pressed at 50MPa, and then sintered at 1500 ºC to make the sample dense. After adding binder, the sintered sample is prepared into a disc with a diameter of about 8mm and a thickness of about 2mm. Then sintered at 800 ºC  for about 1h, the electrodes were prepared by direct welding of platinum paste.( page 12, line344-349)

Special thanks to you for your good comments.

Reviewer 2 Report

Comments and Suggestions for Authors

In this paper, Zr-based nanoparticles were prepared via a micro confined jet mixer reactor operated in continuous mode under supercritical water conditions.. The result and discussion are reasonable and acceptable.However,before publication, there are still some places where caution should be exercised.

 This manuscript needs reasonable revision in the English language. Check the language and revise typos as well as grammatical errors.

  In the part of abstract, full names of BET,OSC,FTIR and TPR are missing. The authors should supplement the missing information to make it convenient for the readers and reviewers to understand.

 In the part of introduction,the reviewer advise that the authors need to explain the novelty of the study in more detail,especially the choice of Y, Ce and Yb as doping phases and their unique advantages among rare earth elements.

In table 1, how did the authors confirm the molar rations of CYZ1,CYZ2 ,CYbZ  and CYYbZ?

Comments on the Quality of English Language

Minor editing of English language required

Author Response

Response to Reviewer 2 Comments

Point 1: This manuscript needs reasonable revision in the English language. Check the language and revise typos as well as grammatical errors.

Response 1: Thank you for your valuable comment. The article has been professionally revised in English to correct grammar and spelling errors

  Point 2: In the part of abstract, full names of BET,OSC,FTIR and TPR are missing. The authors should supplement the missing information to make it convenient for the readers and reviewers to understand.

Response 2: Thank you for your valuable comment. This deficiency has been corrected as follows.

In current study, four kinds of composite oxides consisted of Zr, Y, Ce and Yb were prepared by CHFS method. The effects of different cations and proportions on the structure and properties of Zr-based composite oxides were investigated, and the morphologies of the prepared nanoparticles were characterized by inductively coupled plasma-atomic emission spectroscopy (ICP-AES), scanning electron microscopy- energy dispersive spectrometer (SEM-EDS), X-ray diffraction (XRD), transmittance electron microscopy (TEM), Fourier transform infrared spectroscopy (FTIR) and Raman spectroscopy (RAMAN), the specific surface area of the prepared composite oxide nanoparticles was analyzed by BET test. The oxygen storage and reduction properties of samples were characterized by means of oxygen storage capacity (OSC) test and H2-temperature programmed reduction test (H2-TPR). The ionic conductivity of prepared samples was measured by electrochemical impedance spectroscopy (EIS), and the influence of Y, Ce and Yb elements on ionic conductivity of electrolyte materials was analyzed.( page 3, line109-121)

 Point 3: In the part of introduction,the reviewer advise that the authors need to explain the novelty of the study in more detail, especially the choice of Y, Ce and Yb as doping phases and their unique advantages among rare earth elements.

Response 3: Thank you for your valuable comment. We have added a description of the advantages of the Y, Ce and Yb elements as follows.

In this paper, four elements of Ce, Zr, Y, and Yb are selected to make composite oxides in different proportions. Compared with the oxides of single metal, the composite oxides of multiple metals often have more advantages in performance and application1-2. When a certain amount of Y3+ is introduced into the zirconia lattice to form YSZ, Y3+ cations will replace Zr4+ in the lattice. In order to maintain local electrical neutrality, oxygen vacancies will be generated in the zirconia lattice, which can reduce the repulsion between local O2-, cause large lattice distortion in the ligand space, and release part of the interlayer stress to stabilize the cubic phase C-ZrO23-4. Zacate et al.5 demonstrated that the incorporation of rare earth elements into zirconia resulted in the formation of association defects with oxygen vacancies, thereby establishing a defect association energy between them. The presence of this defect association energy significantly enhanced the binding force between rare earth elements and zirconia crystals. CeO2 has fluorite structure and can be combined with ZrO2 to form cubic CeO2-ZrO2 composite oxide, which is a catalyst with important application value in the field of gas environmental protection. At the same time, the CeO2-ZrO2 composite oxide is also the most common oxygen storage material, because the Ce4+/Ce3+ has a reversible conversion process6-7. In recent years, Yb has emerged and developed rapidly in the fields of optical fiber communication and laser technology. Yb2O3 is the critical material of many high-tech materials such as laser glass and fiber. The electrical conduction in Yb-doped BaZrO3 has been investigated by researchers8.Xu et al.9 introduced association defects by doping rare earth elements Yb into YSZ. They speculated that the defect association between Yb, Y and oxygen defects could make the crystal more stable, and the doped elements were not easy to lose from the crystal. ( page 1, line35- page 2,line57)

Point 4:In table 1, how did the authors confirm the molar rations of CYZ1,CYZ2 ,CYbZ  and CYYbZ?

Response 4: Thank you for your valuable comment. In the revised paper, we added SEM-EDS and ICP-AES tests to characterize the composition and content of elements on the surface of materials.

SEM-EDS energy spectrum (Figure 2) and ICP-AES were used to characterize and analyze the chemical composition and metal element content of the prepared samples. The four samples of CYZ1, CYZ2, CYbZ and CYYbZ did not appear impurity peaks, and the purity was very high, and the C element was produced by the carbon element contained in the conductive adhesive during the preparation process. The stoichiometric ratio of metal elements in the four samples obtained by analysis is consistent with the initial set ratio, and slightly deviates from the initial set ratio. This deviation is caused by the different precipitation kinetics of different precursors, and precipitation is generated in a short time (only a few seconds), and the amount of cation with greater solubility enters the oxide is less (see Table 2).

Figure 2. SEM-EDS spectra of the CYZ1、CYZ2、CYbZ、CYYbZ samples

Table 2. Composition analyses of the obtained CYZ1、CYZ2、CYbZ、CYYbZ samples

(molar ratio)

CYZ1

(Ce: Y: Zr)

CYZ2

(Ce: Y: Zr)

CYbZ

(Ce: Yb: Zr)

CYYbZ

(Ce: Y: Yb: Zr)

EDS

0.94: 1.26: 14.69

14.43: 3.24: 1.98

18.02: 5.04: 2.71

15.76: 1.92: 2.28: 2.13

ICP-AES

1.18: 1.53: 16.07

15.39: 3.76: 2.24

16.81: 4.91: 2.53

18.73: 2.19: 2.71: 2.59

( page 5, line 201- page 6,line 214)

Special thanks to you for your good comments.

Round 2

Reviewer 2 Report

Comments and Suggestions for Authors

accept